# Clinical Impact of a Protocol Involving Cone-Beam CT (CBCT), Fusion Imaging and Ablation Volume Prediction in Percutaneous Image-Guided Microwave Ablation in Patients with Hepatocellular Carcinoma Unsuitable for Standard Ultrasound (US) Guidance

**DOI:** 10.3390/jcm12247598

**Published:** 2023-12-09

**Authors:** Pierpaolo Biondetti, Anna Maria Ierardi, Elena Casiraghi, Alessandro Caruso, Pasquale Grillo, Serena Carriero, Carolina Lanza, Salvatore Alessio Angileri, Angelo Sangiovanni, Massimo Iavarone, Giuseppe Guzzardi, Gianpaolo Carrafiello

**Affiliations:** 1Facoltà di Medicina e Chirurgia, Università degli Studi di Milano, Via Festa del Perdono 7, 20122 Milan, Italyalessandrocars97@gmail.com (A.C.); massimo.iavarone@gmail.com (M.I.); gianpaolo.carrafiello@unimi.it (G.C.); 2Department of Diagnostic and Interventional Radiology, Foundation IRCCS Cà Granda—Ospedale Maggiore Policlinico, Via Francesco Sforza 35, 20122 Milan, Italy; annamaria.ierardi@policlinico.mi.it (A.M.I.); pasquale.grillo@unimi.it (P.G.); alessioangileri@gmail.com (S.A.A.); 3AnacletoLab, Computer Science Department “Giovanni degli Antoni”-DI, Università degli Studi di Milano, 20133 Milan, Italy; elena.casiraghi@unimi.it; 4CINI National Laboratory in Artificial Intelligence and Intelligent Systems, 00185 Rome, Italy; 5Postgraduate School in Radiodiagnostics, Università degli Studi di Milano, 20122 Milan, Italy; serena.carriero@unimi.it; 6Department of Gastroenterology and Hepatology, Foundation IRCCS Cà Granda–Ospedale Maggiore Policlinico, Via Francesco Sforza 35, 20122 Milan, Italy; angelo.sangiovanni@unimi.it; 7Department of Radiology, Unit of Interventional Radiology, Ospedale Maggiore della Carità, Corso Giuseppe Mazzini 18, 28100 Novara, Italy; giuseppe.guzzardi@maggioreosp.novara.it

**Keywords:** cone-beam CT (CBCT), fusion imaging, hepatocellular carcinoma (HCC), microwave ablation

## Abstract

Purpose: to evaluate the clinical impact of a protocol for the image-guided percutaneous microwave ablation (MWA) of hepatocellular carcinoma (HCC) that includes cone-beam computed tomography (CBCT), fusion imaging and ablation volume prediction in patients with hepatocellular carcinoma unsuitable for standard ultrasound (US) guidance. Materials and Methods: this study included all patients with HCC treated with MWA between January 2021 and June 2022 in a tertiary institution. Patients were divided into two groups: Group A, treated following the protocol, and Group B, treated with standard ultrasound (US) guidance. Follow-up images were reviewed to assess residual disease (RD), local tumor progression (LTP) and intrahepatic distant recurrence (IDR). Ablation response at 1 month was also evaluated according to mRECIST. Baseline variables and outcomes were compared between the groups. For 1-month RD, propensity score weighting (PSW) was performed. Results: 80 consecutive patients with 101 HCCs treated with MWA were divided into two groups. Group A had 41 HCCs in 37 patients, and Group B had 60 HCCs in 43 patients. Among all baseline variables, the groups differed regarding their age (mean of 72 years in Group A and 64 years in Group B, respectively), new vs. residual tumor rates (48% Group A vs. 25% Group B, *p* < 0.05) and number of subcapsular tumors (56.7% Group B vs. 31.7% Group A, *p* < 0.05) and perivascular tumors (51.7% Group B vs. 17.1% Group A, *p* < 0.05). The protocol led to repositioning the antenna in 49% of cases. There was a significant difference in 1-month local response between the groups measured as the RD rate and mRECIST outcomes. LTP rates at 3 and 6 months, and IDR rates at 1, 3 and 6 months, showed no significant differences. Among all variables, logistic regression after PSW demonstrated a protective effect of the protocol against 1-month RD. Conclusions: The use of CBCT, fusion imaging and ablation volume prediction during percutaneous MWA of HCCs provided a better 1-month tumor local control. Further studies with a larger population and longer follow-up are needed.

## 1. Introduction

In patients with early-stage HCC, image-guided percutaneous thermal ablation is aimed at eradicating all viable malignant cells of the target tumor [1].

This treatment is influenced by multiple factors that can be broadly divided into tissue-related and technical factors. 

Tissue-specific properties are related to the organ and tumor involved, they affect the efficacy of ablation in achieving local tumor control [2,3] and they cannot be modified by the operator [4,5].

Technical factors, on the other hand, are modifiable by the interventionalist who desires to increase treatment efficacy. 

First, before the procedure, the ablation modality must be chosen; in the liver, microwave ablation (MWA) has shown some advantages over radiofrequency ablation [6].

During the procedure, before choosing the desired ablation power and time, the operator’s efforts are aimed at placing the device in a correct position. The ablation volume should cover the tumor volume, together with an ablative margin of ≥5 mm [7], sparing the non-tumoral surrounding tissue and avoiding damage to non-target structures.

Different modalities are available and are used for imaging guidance, alone or in combination [8]. Ultrasound (US) has the advantage of real-time imaging, but visibility can be limited by deep lesions, large patients or poor tumoral sonographic conspicuity. Computed tomography (CT) has excellent spatial resolution but implies radiation and lacks real-time assessment.

Cone-beam CT (CBCT) is an imaging technology consisting of a rotating C-arm equipped with a flat panel detector that provides volumetric CT datasets. This arm rotates around the patient, who lays on the angiographic table, and it is carried out before, during or after the interventional procedures [9]. CBCT systems can be used together with dedicated software for ablation volume prediction and fusion imaging. 

A small number of studies have described CBCT fusion imaging in liver ablation; these have mostly been retrospective studies, with CBCT performed after ablation [10,11]. When CBCT was acquired before ablation, three studies were performed in patients under general anesthesia [12,13,14] and only one in patients under moderate/deep sedation [15].

The aim of this study is the evaluation of a protocol for image-guided percutaneous thermal ablation that involves the use of CBCT, fusion imaging and a software for ablation volume prediction. The main aim is to evaluate local tumor control in patients with HCCs unsuitable for standard ultrasound (US) guidance (poor visibility) or a laparoscopic approach. 

## 2. Materials and Methods

Between January 2021 and June 2022, 111 patients with 135 focal liver lesions treated with percutaneous image-guided MWA in a tertiary center were retrospectively evaluated.

Inclusion criteria were age ≥18 years; solitary HCC measuring ≤3.5 cm or ≤3 HCC lesions each measuring ≤3.0 cm; no radiologic evidence of vascular invasion or extra-hepatic disease; Child–Pugh grade A or B; and availability of ≥1 month of radiological follow-up. Exclusion criteria were ablation performed with embolization (combination therapy); treated tumor number < total tumor number; or pre-procedural CT/MR not available in the institutional Picture Archiving and Communication System (PACS). 

The final study population was composed of 80 patients with 101 HCCs (Figure 1) and patients were divided into two groups based on the possibility of approaching the lesion by US guidance or not: Group A (37 patients with 41 HCCs), treated using a protocol involving CBCT, fusion imaging and ablation volume prediction; Group B (43 patients with 60 HCCs), treated with standard US guidance. 

For each patient, the following data were registered: sex; age; comorbidities; cause of chronic liver disease; cirrhosis; Child–Pugh score; blood alpha fetoprotein (AFP); tumor number; and previous liver treatments. For each tumor, the following features were assessed: size (maximal axial diameter), location (Couinaud segment), US visibility (visible, poorly visible or non-visible), position (subcapsular, pericholecystic, close to gastrointestinal structures, perivascular) and nature (new versus residual tumor from previous treatments).

On pre-procedural CT/MR, a tumor was defined as being subcapsular when it was located ≤5 mm from the capsule [16], pericholecystic when it was ≤5 mm from the gallbladder, close to gastrointestinal structures when it was ≤5 mm from them [17,18,19] and perivascular when it was in contact with a vessel of ≥3 mm [19].

### 2.1. Procedure

The indication for treatment was given by a multidisciplinary board according to standard criteria. All procedures were performed after informed consent was obtained and antibiotic prophylaxis was given, with the assistance of an anesthesiologist experienced in moderate/deep sedation. 

Treatment protocol: Group A—CBCT, fusion imaging and ablation volume prediction.

The protocol’s steps used to treat Group A patients are represented in Figure 2. First, pre-procedural CT/MR images were loaded to a workstation (XtraVision, Philips Image-Guided Therapy) and the tumor was segmented.

With each patient supine on the angiographic table (Azurion Clarity, Philips Medical Systems, Best, The Netherlands), after local anesthesia, a straight 13.5-gauge microwave antenna (Emprint Microwave Ablation System, Medtronic and Covidien, Boulder, CO, USA) was positioned under US guidance (Epiq 5, Philips Medical Systems, Best, The Netherlands) using one or a combination of the following: B-mode, contrast-enhanced US (CEUS), and/or US fusion imaging (using pre-procedural CT/MR).

Once the antenna was judged to be in place, an unenhanced CBCT was performed with the patient possibly holding their breath to produce as similar as possible conditions to the pre-procedural CT/MRI. Each CBCT acquisition (XperCT, Philips Image-Guided Therapy) consisted of 312 X-ray projections acquired throughout a 240-degree rotation of the C-arm around the patient in 5.2 s (120 kV), with an “open-arc” trajectory [15,20]. Data were transferred to the workstation and resulted in a volumetric reconstruction.

After having been transferred to the workstation, the unenhanced CBCT images, showing the antenna position, were registered to the pre-procedural CT/MR images, showing the tumor. Software (Allura Xper FD20 Philips Medical Systems, Netherlands), was then used to place a “virtual antenna” exactly over the real one, at the tip of which, after having selected the desired ablation power and time, the predicted ablation volume was displayed. The aim was coverage of the tumor with a 5 mm safety margin, avoiding non-target structures (Figure 3).

If the antenna resulted in a suboptimal position, this was changed under US guidance and another CBCT was performed; this step was repeated until a satisfactory result was obtained. Ablation was performed under US surveillance. Immediate post-procedural US and/or unenhanced CBCT was performed in order to detect early complications. [21]

For Group A patients, the following data were collected: number of CBCTs and antenna repositioning attempts, type of repositioning, fusion quality and radiation exposure (dose–area product (DAP), mGy·cm^2^.

For all patients, complications were classified according to the CIRSE classification [22].

### 2.2. Follow-Up

Radiological follow-up consisted of contrast enhanced CT/MRI 1 month after the procedure, then at 3 months and every 3–4 months thereafter (Figure 4). In the case of tumor detection during follow-up, the indication for a new treatment was discussed in a multidisciplinary setting. Patients undergoing new treatments were censored.

### 2.3. Outcomes

The primary end-point of this study was to evaluate local tumor control, measured as residual disease (RD) or local tumor progression (LTP). RD was the presence of residual viable tumor at the ablative margin on 1-month follow-up imaging, whereas LTP referred to a tumor being detected at the edge of the ablation zone after ≥1 month of follow-up [23].

The secondary end-point of this study was the appearance of a new tumor, distant from the treated area, was defined as intrahepatic distant recurrence (IDR) [23].

RD rates at 1 month, together with LTP rates at 3 and 6 months after the procedure, were calculated per tumor and per patient. IDR rates were calculated per patient at 1, 3 and 6 months [24].

Each session corresponded to one treatment.

Ablation response at 1 month was also evaluated according to the modified response evaluation criteria in solid tumors (mRECIST), as shown in Table 1 [25].

### 2.4. Statistical Analysis

Quantitative variables were expressed as the mean ± standard deviation, categorical variables were defined as specific counts and/or proportions.

Comparison of variables between Groups A and B was performed by Wilcoxon rank sum test for continuous variables and by χ^2^ test or Fisher’s test for categorical variables. Collinearity detection was performed by calculating the generalized variance inflation factor (GVIF) [26]; a GVIF >5 was observed for the following variables: pericholecystic, adjacent to GI, obesity, COPD, cardiopathy, and hemophilia. 

For the outcome in terms of 1-month RD, the available cohorts were adjusted by applying an inverse probability weighting approach [27] by computing inverse propensity score weights, under the ATE estimand [28,29,30]. Variables included in the propensity score model were age, sex, AFP, Child–Pugh grade, tumor number, tumor size, tumor US visibility, tumor nature and previous treatments. The standardized mean difference between the adjusted variables was checked to ensure it was <0.1. The adjusted datasets were used to infer relationships between the variables and the outcome of 1-month RD by weighted logistic regression. The odds ratios were pooled by Rubin’s rule to obtain a final, pooled estimate across all of the amputations. All statistical analyses were performed using R 3.0.2. (R Foundation for Statistical Computing, Vienna, Austria). A *p*-value < 0.05 was considered statistically significant.

## 3. Results

### 3.1. Baseline Population Features 

Between 2021 and 06/2022, 80 patients with 101 HCCs were divided in two groups: Group A (37 patients with 41 HCCs), treated using a protocol involving CBCT, fusion imaging and ablation volume prediction; and Group B (43 patients with 60 HCCs), treated with standard US guidance. The baseline characteristics of the included patients are shown in Table 2. Age was higher in Group A than Group B (mean of 72 and 64 years, respectively, *p*-value < 0.003). Group A had more targets, namely residual disease from prior treatments (48.8% vs. 25%, *p* < 0.025), while Group B had more subcapsular (56.7% vs. 31.7%, *p* < 0.024) and perivascular (51.7% vs. 17.1%, *p* < 0.002) tumors. There were no differences between the groups regarding the other variables.

### 3.2. Protocol-Related Data

The median number of intra-procedural CBCT performed in Group A was 2 (range 1–7); immediate post-procedural CBCT for assessment of early complications was performed in 13.5% patients (n = 5/37). In Group A, the microwave antenna was repositioned based on the information given by the protocol in 48.8% of tumors (n = 20/41) a median number of 2 times (range 1–6), partially in 43.9% (n = 18/41) of tumors and totally in 4.9% (n = 2/41) of tumors.

Registration between pre-procedural CT/MR and intraprocedural CBCT was judged to be optimal in 61% (n = 25/41), satisfying in 26.8% (n = 11/41) and bad in 12.2% (n = 5/41) of tumors.

The mean procedural time was 46 min (range 21–110) and the mean dose was 47.5 Gy·cm^2^. 

### 3.3. Complications

In Group A, minor complications were registered in 2/37 patients (5.4%), both Grade I (one episode of nausea and one of intense pain). In Group B there were two minor complications (4.6%, pain) and one major Grade III complication (2.3%): hemoperitoneum occurred in one hemophilic patient after the ablation of three tumors with no active bleeding on CT; therefore, it was successfully managed conservatively.

During the follow-up, one case of portal vein thrombosis was observed in one Group A patient with progression of intrahepatic disease, and there was one case of an arterioportal fistula in Group B, both at 1 month

There were no procedure-related deaths.

### 3.4. Outcome

Follow-up at 1, 3 and 6 months in Groups A and B was available in 100%, 57% and 38%, and in 100%, 49% and 35% of baseline cases, respectively.

#### 3.4.1. Ablation Response at 1 Month According to mRECIST

Table 3 reports the ablation response at 1 month according to the mRECIST criteria. The local ablation response was different between the groups (*p* < 0.05).

As there was no difference in the appearance of new intrahepatic lesions, the overall ablation response rates were different between the groups (*p* < 0.05). 

#### 3.4.2. Tumor Response

Table 4 reports local ablation responses during follow-up, calculated per tumor.

The RD rates at 1-month were lower in Group A (7.3% in Group A and 36.7% in Group B; *p* < 0.05); LTP rates at 3 and 6 months were not significantly different.

#### 3.4.3. Patient Response

Local and distant ablation responses during follow-up, calculated per patient, are reported in Table 5.

The RD rate at 1 month was lower in Group A (8.1% of patients in Group A and 46.5% in Group B; *p* < 0.05), while no difference was observed for LTP or IDR rates.

#### 3.4.4. Tumor-Level Analysis: Propensity Score Weighting—1-Month RD 

Figure 5 illustrates the covariate balance calculated on a tumor basis, before and after the application of the inverse probability weighting approach.

Figure 6 illustrates the odds ratio for developing residual disease at 1 month, obtained after weighted logistic regression, on a tumor-level analysis. Logistic regression detected a unique significant association (*p* < 0.05) between Group A and outcome, with the resulting OR and CI suggesting a protective effect of the protocol.

## 4. Discussion

Given the availability of multiple imaging guidance modalities and technologies, selecting the best tools and their combination is important.

Our study had the aim of evaluating the clinical impact of a protocol for image-guided percutaneous MWA of HCC that included CBCT, fusion imaging and ablation volume prediction in patients with hepatocellular carcinoma unsuitable for standard US guidance (in Group A) due to the poor visibility of the lesion. 

We found that patients with HCC treated with percutaneous MWA with a protocol involving CBCT fusion imaging and ablation volume prediction had better 1-month local tumor control, as defined by mRECIST, and better 1-month RD rates than patients treated with US guidance only. There were no differences in 3- and 6-month LTP rates or in IDR rates.

Real-time US guidance is widely appreciated, but tumors may have poor sonographic conspicuity, and pseudolesions can be mistaken for targets [31,32,33]. The fusion of US with CT/MR was developed [32,33], but mistargeting still occurs [34,35]. 

CBCT allows an operator-independent anatomical evaluation.

The protocol evaluated in this study was used for targeting, intraprocedural modification and the prediction of technical adequacy.

Some retrospective studies with a limited population have described CBCT fusion imaging, but CBCTs were performed after ablation for tumor coverage assessment [10,11,34]. 

In this present study, a protocol for image-guided percutaneous thermal ablation that involves the use of CBCT, fusion imaging and software for ablation volume prediction, has been proposed in patients with HCC that is unsuitable for standard ultrasound (US) guidance (poor visibility) or a laparoscopic approach. 

Few authors have used CBCT for targeting and intraprocedural modification, but their procedures were performed with general anesthesia [12,13,14]. This certainly allows for precise targeting, but it may exclude some patients from treatment and requires longer time. In our institution, percutaneous ablations are generally performed with moderate–deep sedation, and we tested a protocol which would not have required patient exclusion or schedule modification. To the best of our knowledge, only one study has evaluated CBCT fusion imaging and ablation volume prediction in patients with liver tumors under moderate–deep sedation: Floridi et al. fused contrast-enhanced CBCT images acquired before the procedure to unenhanced CBCTs performed after US-guided device positioning in 15 patients [15]. In that study, a mean number of 4.9 (range 4–7) CBCTs/tumor and a mean DAP of 67.3 mGy·cm^2^ were reported, while we registered a mean of 2.1 (range 1–7) CBCTs/tumor and a mean DAP of 47.5 Gy·cm^2^. This is more in line with the median DAP of 43.8 Gy·cm^2^ reported by Abdel-Rehim et al. [11].

In this study the antenna was repositioned based on the information given by the protocol in 48.8% of tumors, which is slightly lower than the rate reported by Floridi et al. of 73.3% of cases [15].

Regarding the outcome, most studies are limited by their small populations, short duration of follow-up and heterogeneity of tumor histologies; comparisons are limited by the heterogeneity of techniques and outcome measures.

Using CBCT prior to ablation for targeting by virtual navigation led to complete RFA in three patients and to a primary efficacy of 79% (53/67 RFAs or MWAs) [12,13]. CBCT performed before ablation and fused with US for guidance led to a complete response at 1 and 3 months in six patients [14]. Floridi at al. observed a complete response at 1 month with 14/15 ablations, and the remaining patient was treated with debulking intent [15].

In this study, the protocol had a significant impact on local tumor control at 1 month, as demonstrated by RD rates and mRECIST outcomes. A protective effect of the protocol was confirmed by weighted logistic regression analysis, executed after inverse probability weighting.

Our fusion quality was optimal in 61% of tumors, satisfying in 26.8% and bad in 12.2%. This was in line with the results of Abdel-Rehim et al., who reported excellent, good and bad quality in 57%, 17% and 26% of cases, respectively [11].

In the future, the additional use of CBCT, fusion imaging and software for ablation volume prediction in patients with HCCs unsuitable for standard ultrasound (US) guidance (poor visibility) or a laparoscopic approach could be proposed as an alternative protocol to increase the cohort of patients eligible to receive MWA treatment. 

This study has some limitations. The population size was limited, impeding the application of propensity score matching and subgroup analysis. Follow-up was relatively short and the populations available for analysis after the first month were limited.

## 5. Conclusions

In our experience, the use of CBCT fusion imaging and ablation volume prediction during percutaneous MWA of HCC provided better tumor local control at 1 month. No significant impact was observed for local or distant recurrence after the first month. 

Further studies with a larger population and longer follow-up time are needed to confirm the results and identify subgroups of patients who could receive additional benefit from the protocol.

## Figures and Tables

**Figure 1 jcm-12-07598-f001:**
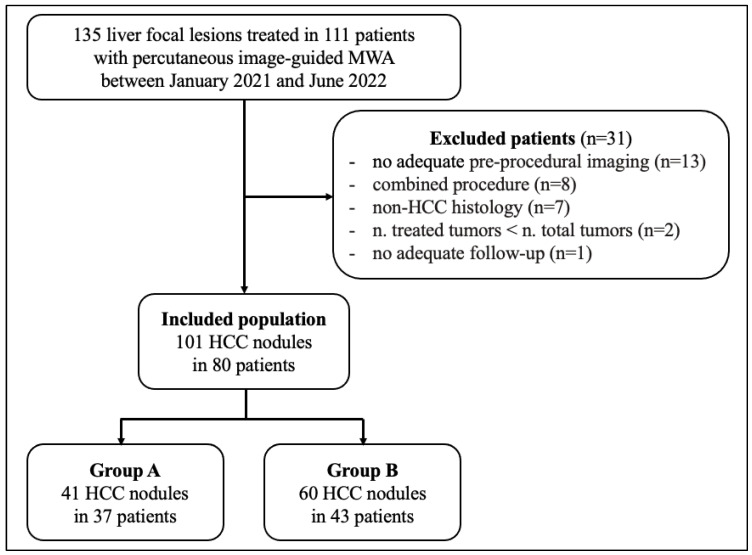
Flow diagram of the study. Abbreviations: HCC, hepatocellular carcinoma; MWA, microwave ablation.

**Figure 2 jcm-12-07598-f002:**
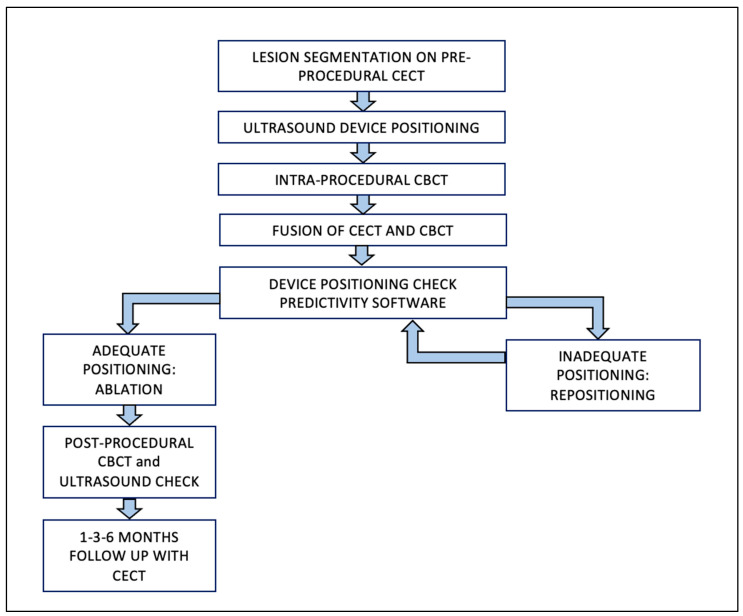
Group A—Procedural Workflow. Abbreviations: CBCT, cone-beam computed tomography; CECT, contrast-enhanced computed tomography.

**Figure 3 jcm-12-07598-f003:**
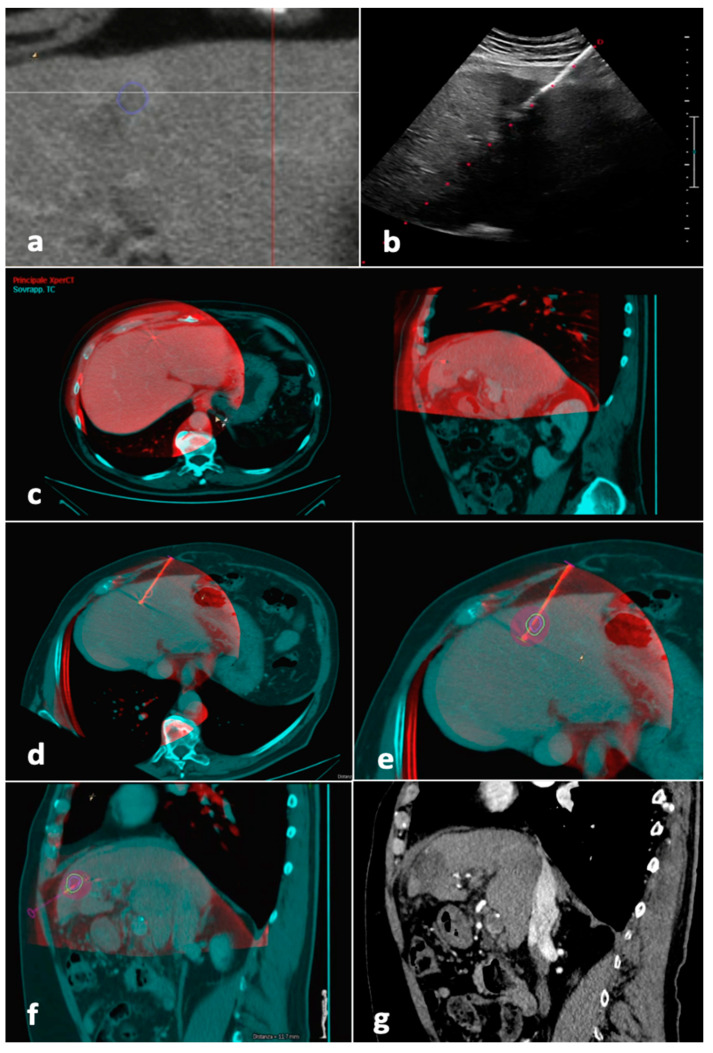
Tumor segmentation, fusion imaging and ablation volume prediction. The tumor, located in the left liver lobe, was semi-automatically segmented (blue line) before the procedure on CT images (**a**). The microwave antenna was positioned into the tumor with US guidance (**b**). When the operator judged the antenna to be in place, CBCT was performed with the patient possibly holding their breath. Manual fusion between pre-procedural CT images showing the tumor (blue) and intraprocedural unenhanced CBCT images demonstrating the microwave antenna (red) was performed, with fused images showing axial and sagittal planes (**c**). On the fused images, a “virtual antenna” was positioned over the microwave antenna (**d**). The predicted ablation volume was generated based on ablation time and power selected (purple), which in this case, completely covers the target tumor (blue line) and the 5 mm safety margin (green line) in the axial (**e**) and sagittal (**f**) planes. (**g**) Follow-up sagittal image in the arterial phase obtained 1 month after procedure shows complete response. Abbreviations—CBCT, cone-beam CT; US, ultrasound.

**Figure 4 jcm-12-07598-f004:**
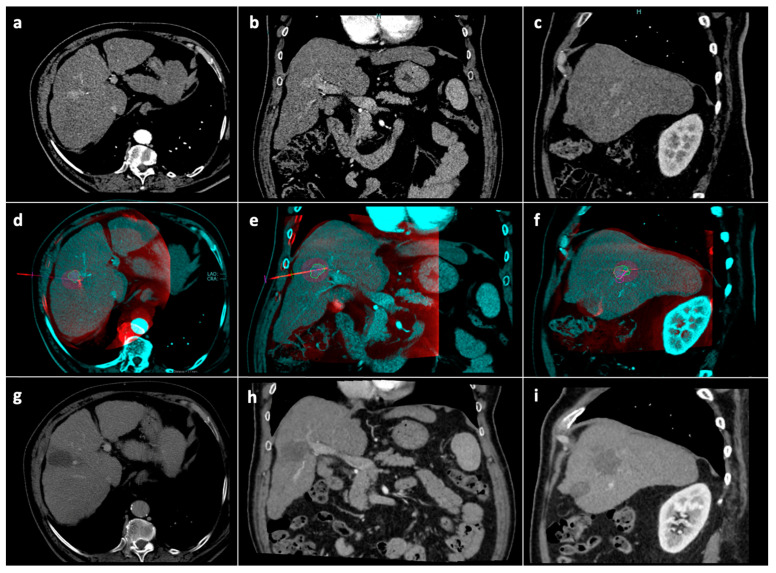
Tumor segmentation, ablation volume prediction and follow-up control. Pre-procedural CT arterial-phase images in the axial (**a**), coronal (**b**) and sagittal (**c**) planes. Intraprocedural fusion between preprocedural CT images showing the tumor (blue) and intraprocedural unenhanced CBCT images demonstrating the microwave antenna position (red) in the axial (**d**), coronal (**e**), and sagittal (**f**) planes; on the fused images, the volume (blue line) with a 5 mm safety margin (green line) is covered by the virtual predicted ablation volume (purple), which has been generated based on the antenna position and on the chosen ablation power and time. Follow-up CT images at 1-month in the axial (**g**), coronal (**h**) and sagittal (**i**) planes showing complete response. Abbreviations: CBCT: cone-beam computed tomography; CT computed tomography.

**Figure 5 jcm-12-07598-f005:**
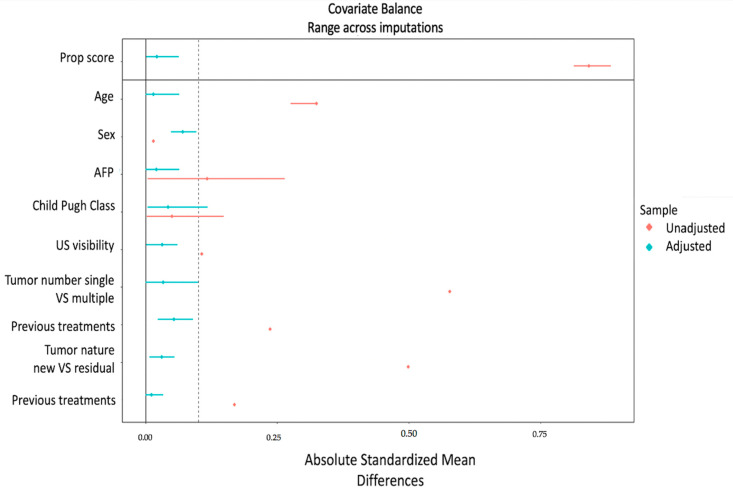
Love plot—Inverse probability weighting for tumors. The absolute standardized mean difference (x-axis) is shown for every covariate (y-axis) and the overall propensity score. Red dots and lines indicate the (pooled) original covariate balance and its CI, and blue dots and lines indicate the (pooled) covariate balance and its CI after weighting. To evaluate the balance of variables used for weighting, the standardized mean difference was checked to determine if values were lower than 0.1.

**Figure 6 jcm-12-07598-f006:**
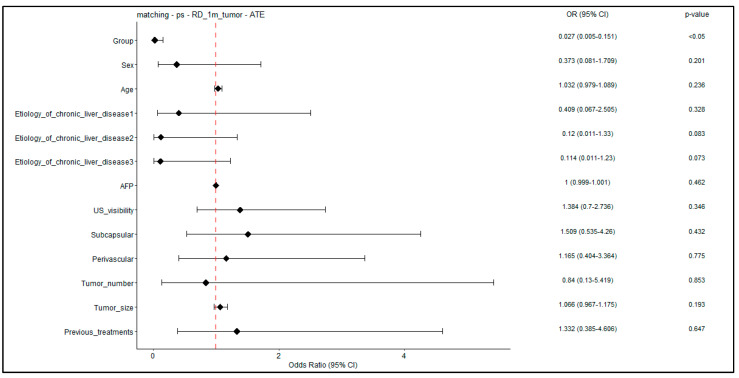
Forest plot–weighted logistic regression for RD at 1 month; tumor-level analysis. Forest plot showing odds ratios (and *p*-values) obtained on the adjusted patient cohort for the 1-month RD outcome. In the logistic regression, the predictor “Group” is a binary variable indicating novel treatment (Group A).

**Table 1 jcm-12-07598-t001:** Response assessment in mRECIST: possible combination of tumor responses in target lesions and in the appearance of new lesions, with resulting corresponding overall response.

Target Lesions	New Lesions	Overall Response
CR	No	CR
PR	No	PR
SD	No	SD
PD	Yes or No	PD
Any	Yes	PD

Target lesion response was classified in four categories: complete response (CR) was the disappearance of arterial enhancement in all target tumors; partial response (PR) was a ≥30% decrease and progressive disease (PD) was a ≥20% increase in the sum of diameters of viable target lesions compared to baseline; cases that did not qualify for either PR or PD were classified as stable disease (SD). New lesions were newly detected nodules distant from ablation zone. The overall response was the result of combined assessment of target and new lesions. Abbreviations: CR, complete response; mRECIST, modified response evaluation criteria in solid tumors; PD, progressive disease; PR, partial response; SD, stable disease.

**Table 2 jcm-12-07598-t002:** Baseline patient- and tumor-related features of the included study population.

Variable	Overall (101 HCCs, 80 pts)	Group A (41 HCCs, 37 pts)	Group B (60 HCCs, 43 pts)	*p*-Value
**Age at enrollment (y; mean, range) ^a^**	68 (38–88)	72 (46–88)	64 (38–87)	0.002
**Male sex (%) ^a^**	61 (76.2)	28 (75.7)	33 (76.7)	1
**Liver cirrhosis (%) ^a^**	79 (98.8)	36 (97.3)	43 (100.0)	0.94
**Etiology of chronic liver disease ^a^**				0.247
Alcohol	9 (11.2)	5 (13.9)	4 (9.5)	
Cryptogenic	1 (1.2)	0 (0.0)	1 (2.4)	
HBV	13 (16.7)	3 (8.3)	10 (23.8)	
HCV	45 (57.7)	21 (58.3)	24 (57.1)	
NASH	9 (11.5)	6 (16.7)	3 (7.1)	
PBC	1 (1.3)	1 (2.8)	0 (0.0)	
**Child–Pugh class (%) ^a^**				0.881
A	72 (90.0)	34 (91.9)	38 (88.4)	
B	8 (10.0)	3 (8.1)	5 (11.6)	
**AFP (ng/mL; mean [IQR]) ^a^**	182.1 [2.6, 27.0]	305.8 [2.9, 24.1]	61.7 [2.2, 27.0]	0.408
**Previous liver treatments ^a^**	47 (58.8)	20 (54.1)	27 (62.8)	0.573
**Tumor number ^a^**				0.195
1	63 (78.8)	32 (86.5)	31 (72.1)	0.195
2–3	17 (21.2)	5 (13.5)	12 (27.9)	
**Tumor size (mm; mean, range) ^b^**	17.9 (10–38)	18.8 (11–35)	17.3 (10–38)	0.205
**Tumor location (Couinaud segment, %) ^b^**				0.129
Segments I–II–III–IV	22 (21.8)	8 (19.5)	14 (23.3)	
Segments V–VI	29 (28.7)	18 (43.9)	11 (18.3)	
Segments VII–VIII	50 (49.5)	15 (36.6)	35 (58.3)	
**US visibility (%) ^b^**				0.227
Visible	63 (62.4)	25 (61.0)	38 (63.3)	
Non- or poorly visible	38 (37.6)	16 (39.0)	22 (36.6)	
**Subcapsular ^b^**	47 (46.5)	13 (31.7)	34 (56.7)	0.023
**Pericholecystic ^b^**	4 (4.0)	2 (4.9)	2 (3.3)	1
**Adjacent to GI ^b^**	3 (3.0)	1 (2.4)	2 (3.3)	1
**Perivascular ^b^**	38 (37.6)	7 (17.1)	31 (51.7)	0.001
**Tumor nature ^b^**				
New	66 (65.3)	21 (51.2)	45 (75.0)	
Residual tumor	35 (34.7)	20 (48.8)	15 (25.0)	0.024
**Comorbidities ^a^**				
Hypertension	34 (42.5)	16 (43.2)	18 (41.9)	1
Diabetes	22 (27.5)	8 (21.6)	14 (32.6)	0.4
Obesity	8 (10.0)	5 (13.5)	3 (7.0)	0.55
COPD	3 (3.8)	3 (8.1)	0 (0.0)	0.189
Cardiopathy	8 (10.0)	4 (10.8)	4 (9.3)	1
Hemophilia	6 (7.5)	3 (8.1)	3 (7.0)	1

Quantitative variables are expressed as mean ± standard deviation, and categorical variables as specific counts and/or proportions. ^a^ Data are calculated per patient. ^b^ Data are calculated per tumor.

**Table 3 jcm-12-07598-t003:** Local and overall ablation responses of Group A and B patients at 1-month follow-up, according to mRECIST criteria.

Outcome at 1 Month (%) *		Group A (n = 37)	Group B (n = 43)	*p*-Value
**Local AR**				0.001
	**CR**	34 (91.9)	23 (53.5)	
	**PR**	2 (5.4)	6 (14.0)	
	**SD**	0 (0.0)	12 (27.9)	
	**PD**	1 (2.7)	2 (4.7)	
**New distant lesions**	Yes	5 (13.5)	5 (11.6)	1
**Overall AR**				0.004
	**CR**	30 (81.1)	22 (51.2)	
	**PR**	1 (2.7)	5 (11.6)	
	**SD**	0 (0.0)	10 (23.3)	
	**PD**	6 (16.2)	6 (14.0)	
	**SD**	0 (0.0)	10 (23.3)	

Abbreviations: CR, complete response; mRECIST, modified response evaluation criteria in solid tumors; PD, progressive disease; PR, partial response; SD, stable disease. * Data are calculated per patient.

**Table 4 jcm-12-07598-t004:** Local responses observed during follow-up in Group A and Group B HCC nodules.

		Group A (n = 41)	Group B (n = 60)	*p*-Value
**Outcome ***	**1-month RD**	3 (7.3)	22 (36.7)	0.002
	**3-month LTP**	1 (2.4)	2 (3.3)	1
	**6-month LTP ^a^**	2 (4.9)	4 (6.6)	0.9

Abbreviations: HCC, hepatocellular carcinoma; LTP, local tumor progression; RD, residual disease. * Data are calculated per tumor here. ^a^ Cumulative results.

**Table 5 jcm-12-07598-t005:** Local and distant responses observed during follow-up, calculated per patient.

			Group A (n = 37)	Group B (n = 43)	*p*-Value
**Outcome ***	**1 month**	**RD**	3 (8.1)	20 (46.5)	<0.001
		**IDR**	5 (13.5)	7 (16.3)	0.763
	**3 months**	**LTP**	1 (2.7)	2 (4.6)	0.965
		**IDR ^a^**	8 (21.6)	12 (27.9)	0.609
	**6 months**	**LTP ^a^**	2 (5.4)	4 (9.3)	0.681
		**IDR ^a^**	9 (24.3)	15 (34.8)	0.334

Abbreviations: IDR, intrahepatic distant recurrence; LTP, local tumor progression; RD, residual disease. * Data are here calculated per patient. ^a^ Cumulative results.

## Data Availability

Data may be provided after a reasoned request.

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
