# Peer review of "Clinical Impact of a Protocol Involving Cone-Beam CT (CBCT), Fusion Imaging and Ablation Volume Prediction in Percutaneous Image-Guided Microwave Ablation in Patients with Hepatocellular Carcinoma Unsuitable for Standard Ultrasound (US) Guidance"

_jcm, 2023, doi:10.3390/jcm12247598_

Round 1

Reviewer 1 Report

Comments and Suggestions for Authors

Manuscript ID: JCM-2662000

Reviewer Report

Thank you for giving me a chance to review this study. It is a good study related to carcinoma Clinical impact of a protocol involving cone-beam CT (CBCT),  fusion imaging and ablation volume prediction in percutaneous image-guided microwave ablation of patients 4 with hepatocellular carcinoma unsuitable with standard 5 ultrasound (US) guidance”

Dear author,

This manuscript is well designed and well-presented the extensive statistics done. The purpose of the study is to evaluate the clinical impact of a protocol for image-guided percutaneous mi-22 crowave ablation (MWA) of hepatocellular carcinoma (HCC) that includes cone-beam computed 23 tomography (CBCT), fusion imaging and ablation volume prediction in patients with hepatocellular 24 carcinoma unsuitable with standard ultrasound (US)-guidance. Results: consecutive patients with 101 HCCs treated with MWA were divided 32 in 2 groups: Group A, treated following the protocol (41 HCCs in 37 patients), and Group B, treated 33 with standard ultrasound (US) guidance (60 HCCs in 43 patients). Among all baseline variables, the 34 groups differed regarding age (mean of 72 Group A and 64 years Group B, respectively), new-vs-35 residual tumor rates (48% Group A vs 25% Group B, p<0.05) and number of subcapsular 36 (56.7%Gruop B vs 31.7% Group A, p<0.05) /perivascular tumors (51.7% Group B vs 17.1% Group A, 37 p<0.05). The protocol led to repositioning the antenna in 49% of cases. There was a significant dif-38 ference in 1-month local response between the groups measured as RD rate and mRECIST out-39 comes. LTP rates at 3 and 6 months, and IDR rates at 1, 3 and 6 months, showed no significant 40 differences. Among all variables, logistic regression after PSW demonstrated a protective effect of 41 the protocol against 1-month RD. Conclusions: the use CBCT, fusion imaging and ablation volume 42 prediction during percutaneous MWA of HCCs provided a better 1-month tumor local control.

Comments

1.     Comment 1: Line no 42 Conclusions: the first letter should start with capital.

 the use of CBCT….It must be: The use of CBCT…..

2.     Comment 2: Line no 368

Institutional review board statement: Ethical approval number and committee name is missing.

3.     Comments: Line no 370 Informed consent: kindly clarify written or oral informed consent and please clarify how do you store the data.

4.     Comments: line no 335 remove Full stop and deep it after the Reference.

5.     Please highlight how this study adds to the current available knowledge.

6.     Please include future recommendations for this study

7.     References: Change the entire paper in accordance with the journal's criteria. All the references has to be corrected according to guidelines of the journal.

Model Referencing style

Staines, H.M.; Kirwan, D.E.; Clark, D.J.; Adams, E.R.; Augustin, Y.; Byrne, R.L.; Cocozza, M.; Cubas-Atienzar, A.I.; Cuevas, L.E.; Cusinato, M.; et al. IgG Seroconversion and Pathophysiology in Severe Acute Respiratory Syndrome Coronavirus 2 Infection. Emerg. Infect. Dis. 2021, 27, 85–91.

An excellent study done by authors.

Author Response

Reviewer Report

Thank you for giving me a chance to review this study. It is a good study related to carcinoma “Clinical impact of a protocol involving cone-beam CT (CBCT), fusion imaging and ablation volume prediction in percutaneous image-guided microwave ablation of patients 4 with hepatocellular carcinoma unsuitable with standard 5 ultrasound (US) guidance”

Dear author,

This manuscript is well designed and well-presented the extensive statistics done. The purpose of the study is to evaluate the clinical impact of a protocol for image-guided percutaneous mi-22 crowave ablation (MWA) of hepatocellular carcinoma (HCC) that includes cone-beam computed 23 tomography (CBCT), fusion imaging and ablation volume prediction in patients with hepatocellular 24 carcinoma unsuitable with standard ultrasound (US)-guidance. Results: consecutive patients with 101 HCCs treated with MWA were divided 32 in 2 groups: Group A, treated following the protocol (41 HCCs in 37 patients), and Group B, treated 33 with standard ultrasound (US) guidance (60 HCCs in 43 patients). Among all baseline variables, the 34 groups differed regarding age (mean of 72 Group A and 64 years Group B, respectively), new-vs-35 residual tumor rates (48% Group A vs 25% Group B, p<0.05) and number of subcapsular 36 (56.7%Gruop B vs 31.7% Group A, p<0.05) /perivascular tumors (51.7% Group B vs 17.1% Group A, 37 p<0.05). The protocol led to repositioning the antenna in 49% of cases. There was a significant dif-38 ference in 1-month local response between the groups measured as RD rate and mRECIST out-39 comes. LTP rates at 3 and 6 months, and IDR rates at 1, 3 and 6 months, showed no significant 40 differences. Among all variables, logistic regression after PSW demonstrated a protective effect of 41 the protocol against 1-month RD. Conclusions: the use CBCT, fusion imaging and ablation volume 42 prediction during percutaneous MWA of HCCs provided a better 1-month tumor local control.

Comments

  1. Comment 1: Line no 42 Conclusions: the first letter should start with capital.

 the use of CBCT….It must be: The use of CBCT…..

Thank you for the comment we have modified as you suggested.

  1. Comment 2: Line no 368

Institutional review board statement: Ethical approval number and committee name is missing.

Thank you for the comment we have added the informations

  1. Comments: Line no 370 Informed consent: kindly clarify written or oral informed consent and please clarify how do you store the data.

Thank you for the comment we have added the informations as you suggested.

  1. Comments: line no 335 remove Full stop and deep it after the Reference.

Thank you for the comment we have modified as you suggested.

  1. Please highlight how this study adds to the current available knowledge.

Thank you for the comment we have done an integration of this discussion part as follow: “Some retrospective studies with limited population have described CBCT fusion imaging but CBCTs were performed after ablation for tumor coverage assessment [10,11,39].

In this present study, a protocol for image-guided percutaneous thermal ablation that involves the use of CBCT, fusion imaging and of a software for ablation volume prediction, has been proposed in patients with HCCs unsuitable with standard ultrasound (US) guidance (poor visibility) or laparoscopic approach. Few authors used CBCT for targeting and intraprocedural modification, but their procedures were performed with general anesthesia [12, 13, 14], which certainly allows a precise targeting, but may exclude some patients from treatment and requires longer time.”

  1. Please include future recommendations for this study

Thank you for the comment we have added this additional information in the discussion as follow: In the future, the additional use of CBCT, fusion imaging and of a software for ablation volume prediction in patients with HCCs unsuitable with standard ultrasound (US) guidance (poor visibility) or laparoscopic approach could be proposed as alternative protocol to increase the cohort of MWA treatment eligible patients.

  1. References: Change the entire paper in accordance with the journal's criteria. All the references has to be corrected according to guidelines of the journal.

Model Referencing style

Staines, H.M.; Kirwan, D.E.; Clark, D.J.; Adams, E.R.; Augustin, Y.; Byrne, R.L.; Cocozza, M.; Cubas-Atienzar, A.I.; Cuevas, L.E.; Cusinato, M.; et al. IgG Seroconversion and Pathophysiology in Severe Acute Respiratory Syndrome Coronavirus 2 Infection. Emerg. Infect. Dis. 202127, 85–91.

 Thank you for the comment we have modified as suggested.

An excellent study done by authors.

Thank you very much for the comments and the suggestions.

Reviewer 2 Report

Comments and Suggestions for Authors

In the present study, the authors aimed to evaluate local tumor control in patients with HCC treated with an image-guided percutaneous thermal ablation protocol, using a combination of CBCT, fusion imaging, and ablation volume prediction software, compared to patients treated using only the US-guidance standard.

The manuscript is interesting, but some clarifications and revisions are needed:

- please clarify the design of the study; the evaluation of patients was prospective or retrospective;

- in materials and methods section, line 93-94, please clarify how the patients were divided into the two groups (randomly?). In line 81, you define patients treated with the CBCT + fusion imaging + software protocol as unsuitable with standard ultrasound; please explain what " unsuitable " means (poor visibility of the lesion?).

- in line 159, please correct with "classification";

- in line 202, clarify whether the t-test or Wilcoxon rank sum test was performed. However, I have a question for the authors: What test did you use to assess normality? Kolmogorov-Smirnov test or Shapiro-Wilk?

- in line 225, please specify p-value;

- figure 5, increases the font size in the figure;

- line 361, delete the "Patients section"

- line 362: insert the "author contributions";

- line 367: delete "please add"

- line 455: delete the duplicate reference (37 and 38 are the same).

Comments on the Quality of English Language

The article is clear and well-written, and the quality of English is good.

Author Response

In the present study, the authors aimed to evaluate local tumor control in patients with HCC treated with an image-guided percutaneous thermal ablation protocol, using a combination of CBCT, fusion imaging, and ablation volume prediction software, compared to patients treated using only the US-guidance standard.

The manuscript is interesting, but some clarifications and revisions are needed:

- please clarify the design of the study; the evaluation of patients was prospective or retrospective;

Thank you for the comment. We have added the informations as follow: “Between January 2021 and June 2022, 111 patients with 135 focal liver lesions treated with percutaneous image-guided MWA in a tertiary center were retrospective evaluated.”

- in materials and methods section, line 93-94, please clarify how the patients were divided into the two groups (randomly?). In line 81, you define patients treated with the CBCT + fusion imaging + software protocol as unsuitable with standard ultrasound; please explain what " unsuitable " means (poor visibility of the lesion?).

Thank you for the comment, we have modified as follow:

The final study population was composed of 80 patients with 101 HCCs (Figure 1) and it was divided in 2 groups based on the possibility to approach the lesion by US guidance or not: group A (37 patients with 41 HCCs), treated using a protocol involving CBCT, fusion imaging and ablation volume prediction; group B (43 patients with 60 HCCs), treated with standard US-guidance.

The aim of this study is the evaluation of a protocol for image-guided percutaneous thermal ablation that involves the use of CBCT, fusion imaging and of a software for ablation volume prediction, with the main to evaluate the local tumor control in patients with HCCs unsuitable with standard ultrasound (US) guidance (poor visibility) or laparoscopic approach.

- in line 159, please correct with "classification";

Thank you for the comment we have modified as suggested

- in line 202, clarify whether the t-test or Wilcoxon rank sum test was performed. However, I have a question for the authors: What test did you use to assess normality? Kolmogorov-Smirnov test or Shapiro-Wilk?

In line 202 there was a typo, for which we are sorry; in our tests we used Wilcoxon rank-sum test to avoid any normality assumption. Indeed, both Shapiro–Wilks test and the Kolmogorov–Smirnov test obtained p-values lower than 0.01, which made us reject the null hypothesis of normality.

- in line 225, please specify p-value;

We specified all the p-values subsection 3.1. Baseline population features (lines 223-231 of the revised version).

- figure 5, increases the font size in the figure;

Thank you for the comment, we have modified as suggested.

- line 361, delete the "Patients section"

Thank you for the comment we have modified as suggested

- line 362: insert the "author contributions";

Thank you for the comment we have added the informations

- line 367: delete "please add"

Thank you for the comment we have modified as suggested

- line 455: delete the duplicate reference (37 and 38 are the same).

Thank you for the comment we have modified as suggested
